# Applications of Lactic Acid Bacteria and Their Bacteriocins against Food Spoilage Microorganisms and Foodborne Pathogens

**DOI:** 10.3390/molecules26227055

**Published:** 2021-11-22

**Authors:** Mduduzi P. Mokoena, Cornelius A. Omatola, Ademola O. Olaniran

**Affiliations:** 1Department of Biotechnology and Food Science, Durban University of Technology (Steve Biko Campus), P.O. Box 1334, Durban 4000, South Africa; 2Discipline of Microbiology, School of Life Sciences, College of Agriculture, Engineering and Science, University of KwaZulu-Natal (Westville Campus), Private Bag X54001, Durban 4000, South Africa; omatola.ca@ksu.edu.ng (C.A.O.); olanirana@ukzan.ac.za (A.O.O.)

**Keywords:** bacteriocins, foodborne pathogens, lactic acid bacteria

## Abstract

Lactic acid bacteria (LAB) are Gram-positive and catalase-negative microorganisms used to produce fermented foods. They appear morphologically as cocci or rods and they do not form spores. LAB used in food fermentation are from the *Lactobacillus* and *Bifidobacterium* genera and are useful in controlling spoilage and pathogenic microbes, due to the bacteriocins and acids that they produce. Consequently, LAB and their bacteriocins have emerged as viable alternatives to chemical food preservatives, curtesy of their qualified presumption of safety (QPS) status. There is growing interest regarding updated literature on the applications of LAB and their products in food safety, inhibition of the proliferation of food spoilage microbes and foodborne pathogens, and the mitigation of viral infections associated with food, as well as in the development of creative food packaging materials. Therefore, this review explores empirical studies, documenting applications and the extent to which LAB isolates and their bacteriocins have been used in the food industry against food spoilage microorganisms and foodborne pathogens including viruses; as well as to highlight the prospects of their numerous novel applications as components of hurdle technology to provide safe and quality food products.

## 1. Introduction

Lactic acid bacteria (LAB) are Gram-positive and catalase-negative microorganisms used to produce fermented foods. They appear morphologically as cocci or rods and they do not form spores. LAB are industrially used in fermentations to improve both the taste and texture of food and feed [1,2,3,4]. They produce copious quantities of organic acids and other inhibitory substances, including bacteriocins, which keep food spoilage microbes and pathogenic microorganisms in check [2,5]. 

Lactic acid bacteria use carbohydrates as carbon and energy sources without the use of oxygen. They produce peroxidases as protect against damage from oxygen by-products. Homofermentative LAB use carbohydrates to produce only lactic acid, while heterofermentative LAB produce lactic acid and other compounds such as acetic acid or alcohol and carbon dioxide [6,7,8]. Antimicrobial peptides produced by selected LAB species are called bacteriocins. Bacteriocins originating from LAB have attracted great industrial and scientific interests as biocontrol agents due to safely and efficiently preventing deterioration of minimally processed food items with the benefit of shelf life extension and prevention of economic loss [9]. Additionally, they have a qualified presumption of safety (QPS) status and can selectively exert antimicrobial defense against bacterial food pathogens on a nanomolar scale, thus guaranteeing the safety of consumers [10,11]. The QPS signifies generic safety in all possible uses and the evaluation encompasses four cardinal points principle-taxonomy, scientific knowledge, safety profiles, and the expected end usage [12]. *Lactobacillus* species including *Streptococcus thermophiles, Lactococcus lactis,* and some species of *Leuconostoc* and *Pediococcus* have gained QPS status [13]. Currently, LAB isolates mainly from the *Lactobacillus* and *Bifidobacterium* genera and their bacteriocins are industrially used in food preservation [14]. 

LAB are known to prevent growth of pathogens, degrade mycotoxins, and have probiotic capabilities [15,16]. However, there is paucity of literature focusing on the ability of LAB to inhibit foodborne pathogens, especially during the current era where consumers are becoming more health conscious when it comes to their food choices. The new omics technologies enable bioprospecting for LAB strains with robust antimicrobials and present an opportunity to maximize their contribution in food and nutrition settings. Hence, this review aimed at elaborating on the roles of LAB and their products in the food industry; the extent to which they have been used hurdles against food spoilage microorganisms and foodborne pathogens; and to highlight emerging trends regarding their novel applications in the food and nutrition industries.

## 2. Classification and Sources of Lactic Acid Bacteria 

Lactic acid bacteria (LAB) are found in diverse habitats including food and feed, water, soil, and sewage, as well as the oral, respiratory, gastrointestinal, and genital tracts of humans and animals, and wherever carbohydrate substrates are available [17,18]. Lactic acid bacteria are classified into genera comprising *Lactobacillus*, *Lactococcus, Leuconostoc*, *Pediococcus*, *Streptococcus*, *Aerococcus*, *Carnobacterium*, *Enterococcus*, *Tetragenococcus*, *Vagococcus,* and *Weissella* [19]. Of these genera, *Lactobacillus* is the most prominent. They are closely associated with terrestrial and marine animals. They are dominant microorganisms in the human gastrointestinal tract, where they outcompete pathogens and contribute in maintaining the health of the host [18,20]. Fermented vegetables are the source of the next dominant *Lactobacillus* species [18,21]. Probiotic *Enterococcus* and *Bifidobacterium* genera are also sourced from the intestines and excreta of humans and animals [22], while *Leuconostoc* and *Pediococcus* are derived from chilled meat, fermented fruits, and vegetables, including wine [14,23].

In general, LAB are human friendly microbes associated with fermented foods such as sour milk and yoghurt and are thus regarded as probiotics, which are bacteria or yeasts that improve human well-being [4]. These probiotic LAB modulate the immune system and do not cause any antigenic reactions [24]. Most probiotic microorganisms currently used in the food industry for humans belong to either the *Lactobacillus* or *Bifidobacterium* genus. *Bifidobacteria* are Gram-positive, non-motile, non-sporulating, anaerobic, and heterofermentative bacteria with a high G + C content. Members of the genus *Lactobacillus* are also Gram-positive, non-motile, and non-sporulating organisms. However, the latter are acid-tolerant facultative anaerobes, can be either homo- or heterofermentative and have a low G + C content [25].

## 3. Bacteriocins Produced by Lactic Acid Bacteria

Bacteriocins are peptides with antimicrobial activities that are metabolites of microorganisms such as lactic acid bacteria (LAB), produced as weaponry in order to gain competitive advantage in the niche environment, while they are innocuous to the producing strains [14,26]. Bacteriocins have different sizes, activities, and biochemical characteristics [14].

Bacteriocin-producing cultures have been industrially applied to inhibit *Listeria monocytogenes* and *Clostridium* spp. in various fermented meats, vacuum-packaged products, and in vegetable-based foods [2]. The current over-use of artificial chemicals to limit food spoilage organisms poses health risks and has led to bacteriocins being presented as alternatives, in synergy with plant phenolic compounds and other antimicrobial agents [8]. The spread of antibiotic resistance and demand for food products with fewer chemical preservatives necessitates search for new alternatives to avoid the abuse of therapeutic antibiotics [14]. LAB isolated from homemade fermented vegetables produce antibacterial substances against both Gram-positive and Gram-negative common foodborne bacterial pathogens. This broad spectrum of inhibition suggests that the LAB strains have a potential as natural biopreservatives in various food products, and to combat foodborne pathogens.

Bacteriocins are mainly bactericidal, while some are bacteriostatic, rendering them useful in the food and pharmaceutical sectors, especially also where fermentation is undesirable. Bacteriocins are effective against Gram-positive toxigenic and pathogenic bacteria, acting by forming pores in the membranes of target microorganisms [14]. Heterofermentative *Lactobacillus* spp. have been demonstrated to keep out spoilage microbes in cheese processing [27].

### 3.1. Classificatio of Lactic Acid Bacteria Bacteriocins

Classification of bacteriocins is based on their biochemical profiles and the characteristics of the genes that produce them. Originally, four classes were recognized, and were later revised into three (Table 1) [18]. Lantibiotics, which undergo substantial post-translational modification, are designated as Class I bacteriocins and nisin is the representative member [14].

Class II bacteriocins consists of small, non-modified, heat-stable peptides, with leucocin A as an example [28]. They are subdivided into Class IIa and Class IIB [26,29,30]. The peptides in this class best work in pairs and the genes encoding the two peptides are located in the same operon [31].

Class III bacteriocins consists of heat-labile proteins (>30 kDa) [14,32].

### 3.2. Production and Biosynthesis of LAB Bacteriocins

Bacteriocins synthesized in the ribosomes and are only active after post-translational modification. These modifications include, but not limited to thioether cross-links and dehydration of serine and threonine residues [26]. Pathways for the synthesis of lantibiotic bacteriocin have been described elsewhere [33]. Bacteriocin expression is regulated either by external induction factors, usually secreted by the producer strain itself or it can be constitutive while bacteriocin biosynthesis depends on environmental conditions such as temperature and pH [33]. Specific immunity proteins protect bacteriocin-producers from their own bacteriocins either by preventing formation of pores in membranes or dislodging bacteriocins form the membranes [26].

### 3.3. Growth Conditions for Optimum Production of LAB Bbacteriocins

Several studies have shown that de Man, Rogosa, and Sharpe (MRS) growth medium is the best for bacteriocin production, especially after optimization of the temperature and pH parameters. A temperature of 34–35 ℃, pH 6.0 and addition of 4% phenyl acetamide with 2% glucose and 2.3–2.5% NaCl concentrations, without culture aeration, provide best conditions for bacteriocin production over 48 h [34,35,36,37,38]. LAB isolates from fermented meat yielded high amount of bacteriocins in tryptone glucose yeast extract, while supplementation of media doubles the yield obtained from MRS under similar growth conditions. Further, incorporation of cysteine and glycine, 1% glycerol, and 30 g/L pyruvic acid enhanced bacteriocin production [39,40,41,42].

## 4. Impact of Foodborne Pathogens on Human Health

Food is an organic substrate rich in nutrients and is capable of supporting the growth of contaminating microorganisms. Foodborne diseases are a major public health concern globally [43]. Bacteria, in particular, are the causative agents of approximately 60% of hospitalization cases [44]. Staphylococcal foodborne infection remains as one of the most prevalent diseases worldwide, resulting from ingestion of contaminated food by preformed *Staphylococcus aureus* enterotoxin [43]. Three types of bacterial foodborne diseases are intoxications, infections, and toxicoinfections [45], which are now briefly described. Ingestion of food containing preformed bacterial toxin such as toxins produced by *Staphylococcus aureus* and *Clostridium botulinum* causes bacterial intoxication. Foodborne infection results from ingestion of food containing viable bacteria such as Salmonella and Listeria, which grow in the host and cause illness [46]. When bacteria present in food, such as *Clostridium perfringens*, are ingested and later produce a toxin in the host, they result in foodborne toxicoinfections. Mycotoxicoses arise from ingesting food contaminated with mycotoxins produced by some fungal species [43,47]. 

Some of the foodborne pathogens that have been isolated include; *Bacillus cereus*, *Campylobacter* spp., *Clostridium botulinum*, *Clostridium perfringens*, *Cronobacter sakazakii*, *Listeria monocytogenes*, *Salmonella enterica* subsp. Typhi and *Salmonella enterica* subsp. Paratyphi, and other *Salmonella* spp., *Shigella* spp., *Staphylococcus aureus*, *Vibrio* spp., and *Yersinia enterocolitica*, *Providentia alcalifaciens*, and *Aeromonas hydrophila* [44,48]. These pathogens cover a wide spectrum of foods including animal products, fruits, and vegetables. Endospore-forming bacteria such as *Clostridium* spp. are a cause for concern in the food canning industry. Fungi such as *Penicillium expansum*, *Aspergillus* spp., *Fusarium* spp are notorious for causing postharvest diseases of fruits and grains, resulting in mycotoxicoses when such foods are ingested by humans.

## 5. Activity of LAB and Their Bacteriocins against Foodborne Pathogens

Food fermentation by LAB is the oldest food preservation technique, and has received prominence from the recent past decades due to the ability of LAB to produce bacteriocins, which are capable of replacing chemical preservatives in the food industry and of acting as alternatives to antibiotics in medicine [49,50]. Unlike most therapeutic antibiotics and synthetic food additives, bacteriocins are natural proteins synthesized by the indigenous microbiota of foods and the ease with which they are degraded by proteases in the human digestive tract and, also, excreted suggests that they are allied with nutritional safety [14]. Further, the ribosomally-synthesized nature of the peptides implies that their intrinsic characteristics could be improved to enhance their biotechnological or industrial application and activity spectra [14]. The ability of LAB to inhibit human pathogens and food spoilage microorganisms in the food industry have been documented [51,52] (Table 2).

### 5.1. Direct LAB Incorporation and Activity against Foodborne Bacteria

Several studies over the years have demonstrated the efficacy of lactic acid and their extracellular products against foodborne pathogens associated with severe illness in humans, particularly in immunocompromised individuals [51,52,53]. To date, different bacteriocin-producing LAB strains have been characterized (Table 1), with promising results as a biopreserver in different industrial application approaches (Figure 1).

Strains of *Lactobacillus casei* isolated from Iranian traditional yoghurts showed potential activity against enteropathogenic *E. coli* and *Salmonella* spp. [53]. Djadouni and Kihal [19] screened LAB from dairy, meat products and agro-industrial wastes and isolated a LAB strain (LBbb0141) that contained antimicrobial compound with a wide spectrum and was inhibitory to ten indicator Gram-positive and Gram-negative strains. Probiotic bacteria isolated from different brands of traditional yoghurts in Egypt exhibited antimicrobial activity at a concentration of 10^9^ CFU/g in vitro against the tested indicator pathogens [54]. 

In a study by Khandare and Patil [55], the potential application of LAB for biopreservation of perishable meat products was assessed by using protective cultures isolated from *idli* batter (fermented Indian soft rice cakes). Three isolates demonstrated equal antagonistic activity against Gram-positive and Gram-negative foodborne pathogens such as *S*. *aureus*, *E. coli*, *S. enterica* subsp. Typhi, *B cereus*, and *P. aeruginosa*. Two Lactobacillus strains, *L. plantarum* and *L. paraplantarum*, displayed the ability to prevent human gut infection by food borne pathogens such as *Listeria*, *Salmonella,* and *Escherichia* spp., by preventing their adhesion to intestinal human cells [56]. Moreover, gastric acid and bile tolerant LAB and Bifidobacteria that were isolated from healthy infant stools displayed antagonistic activities against various foodborne pathogens. These probiotic strains include *Lactobacillus rhamnosus, L. caasei*, *L. plantarum,* and *Bifodobacterium longum* and *B. bifidum*. Whilst the LAB strains inhibited all pathogens tested through antibacterial secretion, *Bifodobacterium* spp. demonstrated a high level of competitive exclusion against all the pathogens tested [57]. 

Mansilla [58] evaluated LAB isolates as biopreservatives against foodborne pathogens and spoilage microorganisms in fresh fruits and vegetables. Although a low percentage of isolates demonstrated high inhibitory activities against foodborne and spoilage microbes, a high number of *Leuconostoc* strains proved to be good antagonists, with a biocontrol potential. Other inhibitory isolates included *Lactobacillus plantarum*, *Weissella cibaria*, *Lactococcus lactis,* and *Enrerococcus munditii*. 

In a study conducted by Fossi et al. [59] to assess the inhibitory potential of LAB isolated from traditionally produced beer and wine on *Escherichia coli*, *Salmonella enterica* subsp. Typhi, and *Staphylococcus aureus*, all LAB isolates inhibited the growth of the test pathogens, mainly by bacteriocin production. *S. enterica* subsp. Typhi, followed by *E. coli*, were the most susceptible pathogens to the inhibitory activity of the LAB isolates. Meanwhile, LAB isolates demonstrated antagonistic activity against foodborne pathogens during the fermentation and storage of *borde* and *shamita* (Ethiopian fermented beverages), by drastically reducing the average count of test pathogens [60]. The findings of this study suggest that LAB isolates are possible candidates for the formulation of industrial starter cultures that are useful to produce safe and bioprotective products, which in turn can be suitable purveyors of probiotic cultures.

Further, the antibacterial activity of LAB isolated from raw milk, curd, tomato, and *dosa* batter was evaluated against common enteric pathogens. Overall, the LAB isolates displayed remarkable activity against tested Gram-positive and Gram-negative pathogenic strains, suggesting the potential application of LAB isolates as natural biopreservatives in different food products [61]. These findings are consistent with data from an earlier study which supports the potential industrial use of LAB as bioprotective agents against foodborne human pathogens in ready-to-eat fruits and vegetables [62]. 

Biofilm formation is a natural growth pattern of microorganisms. However, biofilms of LAB serve as antagonistic effectors against most foodborne pathogenic and spoilage biofilms that currently portend a significant risk factor in the food industry because of their resistance to various levels of biocides used for cleaning and disinfection [63]. Hence, *Lactobacillus* biofilm formation is of significance in clinical and industrial settings. Jalilsoosd et al. [64] investigated strong biofilm formation by a newly isolated *Lactobacillus plantarum* PA21 against pathogenic and putrefaction microorganisms. In this study, only *Salmonella enterica* showed resistance to the biofilm of PA21. LAB isolates from Brazilian foods inhibited the formation of biofilm by *Escherichia coli* O157:H7, *Listeria monocytogenes*, and *Salmonella enterica* subsp. Typhimurium via cells co-aggregation that precluded bacteriocin production [65]. The study showed that LAB biofilm antagonistic activities against foodborne pathogenic biofilms represent a promising method to control their formation on food industrial surfaces in the future.

### 5.2. Direct Lactic Acid Bacteria Incorporation and Activity against Foodborne Viruses 

Lactic acid bacteria (LAB) are evolving as a novel wave of antagonists against some foodborne viruses (rotaviruses, noroviruses, caliciviruses, and coronaviruses) either through the mediation of their metabolites or competitive inhibition of the viral cycle [66,67]. Although viruses do not replicate in the food, the incorporation of LAB in ingested food can exert an antiviral state in the host [67] or serve as a potential oral adjuvant [68].

In a study conducted by Lange-Starke et al. [69] to assess the inhibitory potential of LAB on human norovirus surrogates, the cell-free supernatant of *Lactobacillus*
*curvatus* strain caused a 1.25 log units higher titer reduction of murine norovirus S99 (MNV) compared to the control at raw sausage corresponding pH values of 5.0 to 6.2 in vitro. Similarly, Aboubakr et al., [70] demonstrated that a culture filtrate of *Lactococcus lactis* subsp. Lactis LM0230 significantly inhibited the human norovirus surrogate. Another study by Kim et al., [68] showed that the exopolysaccharides from *Lactobacillus plantarum* LRCC5310 sourced from a Korean fermented food significantly inhibited the replication of human rotaviruses in vitro and in vivo, thus suggesting the beneficial role of LAB incorporation against gastrointestinal virus in food that essentially constitute vehicle of transmission via the fecal-oral route. 

Martin et al. [71] showed that heat-inactivated Lactobacillus and Pediococcus obtained from nutritional breast milk completely inhibited infection of a cell by HIV-1 employing CXCR4 and R5/X4 as co-receptors. The inhibition was attributed to the binding of LAB peptidoglycans and/or exopolysaccharide moieties to HIV-1, thus preventing the virus from interacting with the infant’s intestine. This technological characteristic showed that LAB incorporation into commercial human breast milk for infant feeding may be beneficial since the protective activity of LAB is not destroyed by the heating [67]. In another study, the cell culture supernatant of *Lactobacillus plantarum* isolated from Thai pigs was able to inhibit the pandemic strain of a coronaviruses-Porcine epidemic diarrhea virus [72]. 

So far, information on the killing ability of LAB against viruses is in the context of immunomodulation of the host immunological response [67] rather than their direct antiviral effects in foods. It has been hypothesized that the presence of LAB in foods indirectly protect consumers from viral infections through the blocking of receptor sites on the host cell and neutralization of the viral infectivity [71] or boosting of the host immune system to counter viral infections following food consumption [67]. Therefore, studies looking at the possible interactions of the LAB and their cell-free supernatant with virus replication steps or with the monolayer cell line may help to further unravel the mechanism of LAB-mediated inhibition of viruses as well as the novelty of their potential antiviral applications.

### 5.3. Bacteriocins in Antimicrobial Packaging Films and Coatings

Active packaging such as vacuum packaging, active scavenging, and modified atmosphere packaging, are currently employed to increase the strength of normal packaging of highly perishable food including fresh produce, fish, and meat [73]. Bacteriocin incorporation in edible coatings or films has been shown to represent a promising alternative for preserving the microbiological safety and sensory properties of foods that are consumed raw or without further cooking [74]. 

Antimicrobial packaging films impregnated with LAB have been allied to shelf-life extension through continuous interaction with the food matrix and improvement of LAB bacteriocin stability as it gradually diffuses antimicrobial peptides into food [73]. According to Balciunas et al. [75], the application of LAB in the packaging system protects against loss of antimicrobial functionality during the latency phase of pathogenic and spoilage microorganisms. Interestingly, the bioactive packaging film has been shown to exerts greater antimicrobial inhibition than most of the modern food packaging technologies [76], a property that when harnessed may overcome current challenges associated with pathogenic microflora and post-process contamination. 

Nisin and pediocins are the two most commercialized bioactive packaging bacteriocins currently used to prevent foodstuffs, particularly meat and cheese, from spoilage organisms and pathogens in the food industry [77]. In a study by Neetoo et al. [78], the nisin coating of a synthetic film on vacuum-packed cold-smoked salmon elicited a significant reduction in the survival rate of *Listeria monocytogenes*. In another study, the impregnation of a biodegradable food packaging film with bacteriocin from *Weissella hellenica* BCC 7293 resulted in 2 to 5 log CFU/cm^2^ reduction of targeted pathogens (*Listeria monocytogenes, Staphylococcus aureus*, *Aeromonas hydrophila*, *Pseudomonas aeruginosa*, *Salmonella enterica* subsp. Typhimurium, and *Escherichia coli*) in *Pangasius bocourti* fish fillets [79]. 

A study of semi-solid cheese by Cao-Hoang et al. [80] documented a 1.1 log cfu/g reduction in *Listeria innocua* counts after 7-days of cheese storage in a nisin coated film of sodium caseinate. In another study, the coating of cheese with galactomannan and nisin resulted in complete growth inhibition of *Listeria monocytogenes* within 7 days at 40 ℃ [71]. More recently, a biodegradable film impregnated with bacteriocin-like substances of *Lactobacillus curvatus* P99 exerted a complete growth inhibition of *Listeria monocytogenes* in sliced Prato cheese for 10 days of storage at 40 ℃ [81]. Similarly, sea bass (*Centropomus undecimalis*) fillets coated with a glycerol film containing *Lactobacillus reuteri* inhibited the growth of aerobic, enterobacterial, and psychrotrophic microorganisms in 2–3 days relative to sea bass fillets without the film. Additionally, the color and texture of the food were improved as a result of fermentation, preserving the matrix structure, and inhibiting oxidation reactions [82].

### 5.4. Lactic Acid Bacterial Bacteriocins Combination with Other Hurdle Technologies

One interesting fact about bacteriocins is that the antimicrobial activity can be improved when combined with other barriers (e.g., chemical additives, high pressure, and heating treatments) to foodborne pathogens. The uniqueness of bacteriocin combination is that the included barrier (s) are needed at a reduced treatment level for optimal antimicrobial activity, in which case, the undesirable effects of the chemical or physical procedures are minimized with cost-saving benefits [76]. The Nisaplin® product, which employs hurdle technology involving nisin, has received the World Health Organization prequalification for use in food industrial applications and is commercially available in about 50 countries in the world [73,83]. 

In a study developed by Narayanan et al. [84], the incorporation of eugenol into polyhydroxybutyrate films and their combination with pediocin synergistically increased the antimicrobial effect of the film against the growth of food pathogenic microflora and spoilage microbes. In another study, the combined use of nisin and the lactoperoxidase system (LPS) exerted a synergistic effect in the control of *Listeria monocytogenes* in skim milk [85]. The combined huddle produced 5.6 log units lower in *Listeria monocytogenes* counts than the control milk just after 24 h at 30 ℃. Further, Zapico et al. [85] showed that when the LPS and nisin were introduced stepwise at a 2 h interval of growth of *Listeria monocytogenes*, the difference in bacterial counts increased by 7.4 log units. 

The combined application of bacteriocins with ethylenediaminetetraacetic acid (EDTA) is one most frequent strategy currently employed in promoting the sensitization of Gram-negative bacteria. EDTA facilitates the disruption of the bacterial outer membrane to enhance the activity of bacteriocin against Gram-negative organisms notably *Salmonella enterica* subsp. Typhimurium, *Enterobacter aerogenes*, *Shigella flexneri*, *Citrobacter freundii*, *Escherichia coli* O157:H7, *Pseudomonas aeruginosa*, and *Arcobacter butzleri* [86]. Interestingly, low input EDTA of 10 to 20 mM is usually sufficient to produce sensitization for bacteriocin action [86].

Emerging reports have shown that high-pressure processing (HPP) treatment does not always inactivate most microorganisms completely due to the protection of microbial cells by the food constituents or their recovery facilitated by the food substrate post-treatment [87]. However, a complementation strategy involving LAB bacteriocin with HPP has been shown to enhance the sensitivity of the pressure-resistant spoilage bacteria and killing of residual strains by bacteriocin [87,88]. Zhao et al. [89] observed a complete inactivation of the growth of yeast, molds, and total aerobic bacteria in cucumber juice drinks after 50 days of storage at 4 ℃ when a high hydrostatic pressure (500 MPa/2 min) with 100 IU/ml nisin treatment was employed.

Moreover, the synergistic effect of bacteriocins after temperature treatments, with time and cost-saving benefits have been documented [73,87]. High or low-temperature treatments may disintegrate the bacterial outer membrane to promote bacteriocins permeabilization in the cell as was reported for nisin activity against *Salmonella enterica* subsp. Typhimurium and *Escherichia coli* at refrigeration temperatures [87]. Additionally, the efficacy of nisin against *L*. *monocytogenes* improved when combined with NaCl [90]. Further, the exertion of anti-Listerial activity by nisin at low pH pinpoints the suitability of its applications in acidic foods.

Studies have shown that plant essential oils such as thymol and carvacrol, are capable of disrupting bacterial cell membranes making them susceptible to bacteriocin through antimicrobial synergy [87]. The combined inhibitory activity of carvacrol and pediocin against *E. coli* O157:H7 confirms the effectiveness of this strategy [91]. Similarly, the dual application of carvacrol or thymol and nisin produced a significant inhibition of *Salmonella enterica* subsp. Enteritidis in sheep meat [92]. Although, the use of essential oils or their derivatives alone in foods is limited due to sensory changes associated with high concentrations required to exert antimicrobial activity [92]. On the other hand, their use with bacteriocin have been shown to reduce the amount of antimicrobial added to foods, thereby preventing possible undesirable sensory changes [91,92]. A study by Moosavy et al. [93] on the effect of *Zataria multiflora* subsp. Boiss essential oil on *Salmonella enterica* subsp. Typhimurium and *Staphylococcus aureus*, showed that the inclusion of nisin significantly reduced the concentration of oil needed for the inhibition of both bacteria.

## 6. Future Prospects of LAB and Bacteriocins in the Food Industry

There has been a global increase in the consumption of fermented foods as these products are generally regarded as safe; and there is a corresponding increase in the application bacteriocins in food preservation. Various fermented products that incorporate live probiotic LAB cultures are now commercially available and can be consumed by people of all ages, and can mitigate lifestyle disorders [94,95]. Metabolic disorders, which are on the increase globally, arise from diet change and lack of physical exercises. Probiotic consumption has recently been found to ameliorate type 2 diabetes mellitus, cardiovascular disease (by lowering cholesterol levels), and obesity [96]. It is expected that improving and optimizing applications of LAB and their bacteriocins, with their creative combinations with other agents will prolong the positive role they are playing in the food industry.

## 7. Conclusions

The rapid global population growth has resulted in food scarcity, and food security is paramount, especially if the food shelf-life is to be prolonged. LAB and their bacteriocins have emerged as great alternatives to chemicals as preservatives, as they inhibit spoilage microorganisms and gastrointestinal pathogens. Since nowadays consumers prefer products with less chemically synthesized preservatives, LAB and their bacteriocins have emerged as viable alternatives curtesy of their QPS status. There is growing body of knowledge regarding the industrial applications of LAB and their products, the versatile role they play as part of various hurdle technologies to inhibit the proliferation of food spoilage microbes and foodborne pathogens, and the mitigation of viral infections associated with food. As a result, there is emergence of creative industrial applications of LAB such as in the development of novel food packaging and coating materials. As the consumption of fermented foods such as yoghurt and probiotic supplements is globally increasing, applications of LAB and their bacteriocins in the food industry are gaining traction, with the potential to be further enhanced. 

## Figures and Tables

**Figure 1 molecules-26-07055-f001:**
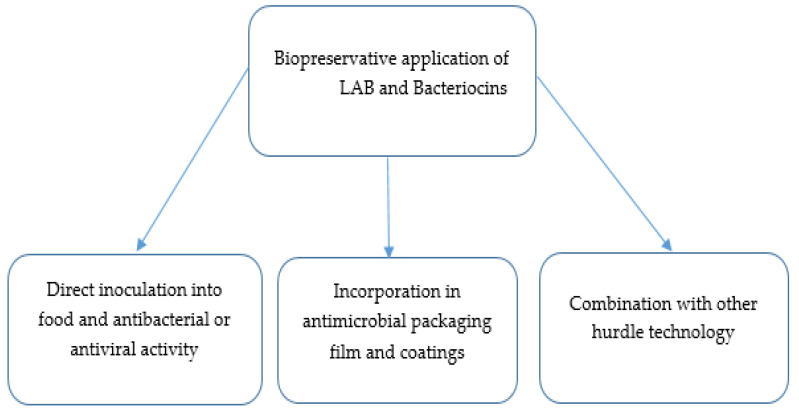
Modes of applications of bacteriocins in the control of foodborne pathogens.

**Table 1 molecules-26-07055-t001:** Classification and properties of LAB bacteriocins.

Classes	Source	Biochenical Profiles	Examples
I	*Lactobacillus lactis* subsp. Lactis	Have lanthionine and methyllanthionine; <5 kDa	Nisin [14,16]
IIa	*Leuconostoc gelidum*	Thermostable, non-modified, cationic, hydrophobic peptides; <10 kDa	Leucocin A [16,26,33]
IIb	*Enterococcus faecium*	Cationic peptide pairs	Enterocin X [28,29]
III	*Lactobacillus helveticus*	Large peptides; heat-labile; >30 kDa	Helveticin J [14,16,28]

**Table 2 molecules-26-07055-t002:** Lactic acid bacteria used against foodborne pathogens in the food industry.

LAB Species	Spectrum of Action	References
*L. casei*	*E. coli*; *Salmonella* spp.	[53]
*L. plantarum*; *L. paraplantarum*	*Listeria* spp.; *Salmonella* spp.; *Escherichia* spp.; *Aeromonas hydrophila*; *B. cereus*; *P. fluorescens*	[51,52,53]
*L. sake*	*Listeria monocytogenes*; *Leuconostoc* spp.; *Pediococcus* spp.	[9]
*Leuconostoc mesenteroides*	*Enterococcus faecalis*; *Listeria monocytogenes*	[9]
*Pediococcus pentosaceous*	*Listeria* spp.; *Clostridium* spp.	[9]
*Enterococcus faecium*	*Listeria monocytogenes*; *Pediococcus* spp.	[9]

## Data Availability

Not applicable.

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
