# Peer review of "Applications of Lactic Acid Bacteria and Their Bacteriocins against Food Spoilage Microorganisms and Foodborne Pathogens"

_molecules, 2021, doi:10.3390/molecules26227055_

Round 1

Reviewer 1 Report

In this review article authors have discussed applications of Lactic Acid Bacteria and their bacteriocins against food spoilage microorganisms and foodborne pathogens. In my opinion the authors have explored the relevant topics exhaustively. The material covered provides a sound overview of necessary technical primer to learn about the field. The language used is clear and arguments are persuasive. I recommend the publication of this manuscript in present form.

Author Response

Errors pertaining to English language and style and spelling have been fixed.

Reviewer 2 Report

The authors discuss the antimicrobial activity of lactic acid bacteria and bacteriocins against food-borne pathogens. Overall, it is thought to provide useful information, and the structure of the thesis is well established.
1. However, it is suggested to provide industrially applied examples of lactic acid bacteria and bacteriocins.
2. In addition, I think it is necessary to mention the principle of beneficial action by acting on pathogens.

Author Response

Errors pertaining to English language and style and spelling have been fixed. Examples of industrially used lactic acid bacteria are captured in Table 1 and elsewhere in the text. In fact, most of the information about uses or applications of LAB pertains to those that are widely used in the food industry. Further, the major mechanism of bacteriocins is pore formation on the membranes of spoilage and pathogenic bacteria.

Reviewer 3 Report

Dear Authors, 

The present study ID:molecules-1459689 entitled "Applications of Lactic Acid Bacteria and Their Bacteriocins Against Food Spoilage Microorganisms and Foodborne Pathogens" written by authors Mduduzi Paul Mokoena, Cornelius Arome Omatola, Ademola Olufolahan Olaniran.

Manuscript deals with the application of lactic acid bacteria and the bactericures produced by them. These substances are important inhibitors of the survival of many other microorganisms, which can be used industrially. The topic of manuscript is undoubtedly interesting. The text is written quite clearly and clearly. However, I think about the entirely new benefits of this study. A large number of reviews of publications on this topic have already been written. Could the authors clearly explain the motivation to write such a focused review and also clearly emphasize the new benefits of this publication? So what's new? I see this as a major drawback to the whole text. The often incorrectly stated names of microorganisms are also quite problematic (many typos, often probably also the ignorance of the authors of the current taxonomy of microorganisms!). This is evident in many places in the text, eg L. 52, L. 243-252, 274-309, L.343, L. 447, ... etc. Indeed, it is astonishing how misspelled the text is. The same is true for the long-obsolete salmonella taxonomy - see L. 221, L.285, 301, 303, .... Furthermore, the taxonomy - Enterobacter sakazakii (see Cronobacter taxonomy), which has been invalid for many years, also appears in the text. Italics are missing in some places in the text - L. 341, etc .; elsewhere, on the other hand, italics is misused - see L. 51, Table 1 (class 1), Table 2 (spp.). A significant problem with the whole text is also a completely poorly written list of used literature! The names of magazines and their abbreviations are incorrectly stated and even in a completely inconsistent form !! In such a state, I would assume that the manuscript does not even reach the opponent.

Minor comments:

1 / L. 108 - wrong character for temperature

2 / L. 207 - could the authors explain the given values ​​with the exponent? The CFU / ml values ​​written in this way look quite strange, apparently a misinterpretation.

3 / L. 220 - "see." - not completely usable in English!

4 / Fig 1 - use a different font than the text!

Author Response

Without a doubt, lactic acid bacteria constitute a wide field of study that interests researchers from food science & nutrition; and biotechnology, to name a few. In this review article, we have discussed applications of Lactic Acid Bacteria and their bacteriocins against food spoilage microorganisms and foodborne pathogens. Hence we limited ourselves to the topic by exploring the relevant subtopics exhaustively in order to provides sound overview of necessary technical information to learn about the field. 

All the grammatical errors regarding spelling and English usage have been fixed, the use of italics for names of organisms has been used consistently. The current taxonomy of Salmonella has been used for the names of these organisms  where they appear in the text, and the refence of the magazine article has been revised.

We thank the reviewer for a robust feedback, from which we have learnt a lot.

Round 2

Reviewer 3 Report

Dear Authors, 

the manuscript has been improved. However, it is important to write citations in accordance with journal requirements and also with general standards (abbreviation form of journal name, etc.). Please, correct them. 

The text include some other errors - eg Tab 1 - "...l(L)actis"

Tab 2 - "spp." write without italics

I recommended unifying the font used in Fig1.

L.393 - as I mentioned in first review process, there is error in some microbes name (eg "Lactis" - write with lowercase letter).